# Effectiveness, Timing and Procedural Aspects of Cognitive Behavioral Therapy after Deep Brain Stimulation for Therapy-Resistant Obsessive Compulsive Disorder: A Systematic Review

**DOI:** 10.3390/jcm9082383

**Published:** 2020-07-26

**Authors:** Meltem Görmezoğlu, Tim Bouwens van der Vlis, Koen Schruers, Linda Ackermans, Mircea Polosan, Albert F.G. Leentjens

**Affiliations:** 1Department of Psychiatry, Ondokuz Mayıs University, 55270 Samsun, Turkey; gormezoglumeltem@gmail.com; 2Department of Neurosurgery, Maastricht University Medical Centre, 6229 Maastricht, The Netherlands; tim.bouwens@mumc.nl (T.B.v.d.V.); linda.ackermans@mumc.nl (L.A.); 3Department of Psychiatry and Neuropsychology, Maastricht University Medical Center, 6229 Maastricht, The Netherlands; koen.schruers@maastrichtuniversity.nl; 4School of Mental Health and Neuroscience, Maastricht University, 6229 Maastricht, The Netherlands; 5Grenoble Institute of Neurosciences, University of Grenoble Alpes, 38058 Grenoble, France; MPolosan@chu-grenoble.fr

**Keywords:** deep brain stimulation, obsessive compulsive disorder, cognitive behavioral therapy

## Abstract

*Background and aim:* Deep brain stimulation (DBS) is an effective treatment for patients with severe therapy-resistant obsessive-compulsive disorder (OCD). After initiating DBS many patients still require medication and/or behavioral therapy to deal with persisting symptoms and habitual behaviors. The clinical practice of administering postoperative cognitive behavioral therapy (CBT) varies widely, and there are no clinical guidelines for this add-on therapy. The aim of this review is to assess the efficacy, timing and procedural aspects of postoperative CBT in OCD patients treated with DBS. *Method:* Systematic review of literature. *Results:* The search yielded 5 original studies, one case series and three reviews. Only two clinical trials have explicitly focused on the effectiveness of CBT added to DBS in patients with therapy-resistant OCD. These two studies both showed effectiveness of CBT. However, they had a distinctly different design, very small sample sizes and different ways of administering the therapy. Therefore, no firm conclusions can be drawn or recommendations made for administering CBT after DBS for therapy-resistant OCD. *Conclusion:* The effectiveness, timing and procedural aspects of CBT added to DBS in therapy-resistant OCD have hardly been studied. Preliminary evidence indicates that CBT has an added effect in OCD patients being treated with DBS. Since the overall treatment effect is the combined result of DBS, medication and CBT, future trials should be designed in such a way that they allow quantification of the effects of these add-on therapies in OCD patients treated with DBS. Only in this way information can be gathered that contributes to the development of an algorithm and clinical guidelines for concomittant therapies to optimize treatment effects in OCD patients being treated with DBS.

## 1. Introduction

Obsessive-compulsive disorder (OCD) is a chronic psychiatric disorder characterized by the presence of obsessions and/or compulsions. Its prevalence varies from 0.8% to 3% in the adult population [1]. The World Health Organization lists OCD as the 11th most common cause of secondary disability, accounting for 2.2% of the total years lived with disability [2]. Cognitive behavioral therapy (CBT), including exposure and response prevention (ERP), as well as pharmacotherapy with serotonergic medication, are the main forms of treatment for OCD. The effectiveness of CBT as treatment for OCD has been established in multiple studies [3,4]. However, its acceptability is limited: as much as 16% to 30% of patients offered CBT drops out during treatment [4,5]. Serotonergic medication, as well as augmentation with atypical antipsychotics, were shown to be effective, and the combination of medication and CBT is even more effective [6,7]. However, a large percentage of OCD patients show only a partial response, or are refractory to psychotherapy and pharmacotherapy. In severe therapy-resistant cases, deep brain stimulation (DBS) may be an option. Although the target and stimulation characteristics may vary across studies and clinics, DBS is generally considered safe and effective for the treatment of therapy-resistant OCD [8]^.^ DBS received approval as treatment for OCD by the European Comission (EC) in 2009, and in the same year it was approved as a Humanitarian Device Exemption by the U.S. Food and Drug Administration (FDA).

After DBS, patients often still need medication, and CBT is often offered because it is considered useful in the treatment of remaining obsessive and compulsive symptoms, in dealing with behavior that has become habitual and persists even when the urge has subsided, and in helping to adjust to the new situation and new expectancies. In addition, CBT provides the patient with new coping styles and problem solving skills that may be important to prevent relapse and contribute to the long-term efficacy of DBS. Whereas guidelines for CBT in OCD have suggested offering CBT after DBS, clinical practice varies widely across institutions and often depends on local possibilities and traditions [9,10]. A more uniform and evidence-based approach may be beneficial for patients.

Up until now, the added effect of CBT to DBS for OCD has not been reviewed. The aim of this systematic review is to assess the literature on efficacy, timing and procedural aspects of postoperative CBT in patients being treated with DBS for therapy-resistant OCD, and to formulate clinical recommendations for future research and for offering CBT after DBS.

## 2. Methods

A systematic review of studies catalogued in PubMed was performed following the Preferred Reporting Items for Systematic Reviews and Meta-Analyses (PRISMA) guidelines (www.prisma-statement.org). The search was done over the full time span up until 17 April 2020. Only papers in English were included. We used the following broad Booleian search strategy: “(deep brain stimulation) AND (obsessive compulsive disorder) AND ((exposure and response prevention) OR (behavioral therapy) OR (cognitive behavioral therapy))”. Given the limited yield of a narrower, exploratory search, all papers that addressed any form of postoperative CBT in patients receiving DBS for therapy-resistant OCD were included. This not only comprised clinical trials, cohort studies, case series and case studies, but also systematic and narrative reviews on DBS for OCD and position papers if they also commented on CBT after DBS. Reference lists of the included studies were checked for additional papers. The Evidence Project risk of bias tool was used to assess the quality of the included studies (not being reviews) [11]. Two authors rated the quality (MG and AL) and discrepancies were resolved by consensus. Due to the very limited yield of our search, no minimum quality score was applied for inclusion. The effectiveness of CBT added to DBS was quantified by looking at the changes in scores on the Yale–Brown Obsessive Compulsive Scale (Y-BOCS) before and after CBT. Although the initial intention was to perform a meta-analysis, this was not possible because of the limited number of included studies that, in addition, used different indication criteria and different forms and ways of delivery of CBT. Timing and procedural aspects of CBT in the studies are reported in a descriptive way.

## 3. Results

### 3.1. Literature Search

The search yielded 181 papers. One additional paper was added after checking the reference lists of included papers [12]. Based on the title and/or abstract, 154 of these were excluded because of the following reasons: animal studies (*n* = 14), not referring to OCD (*n* = 61), not referring to DBS (*n* = 15), not referring to CBT (*n* = 28), not in English (*n* = 10) and other reasons (including the absence of an abstract) (*n* = 17). The remaining 28 papers were read in full. An additional 19 of these were excluded due to the following reasons: not referring to CBT (*n* = 11), not referring to DBS (*n* = 1) and other reasons (*n* = 7). Eventually, 9 papers were included in the review: three randomized controlled trials (RCTs) [13,14,15], one cohort study [16], one case series [12], one qualitative study [17], one systematic review [18] and 2 narrative reviews [19,20]. These reviews were focused on the efficacy of DBS for OCD and not on the efficacy of CBT after DBS for OCD. Two of the included papers were based on the same RCT [13,14] (see Figure 1 for a PRISMA flowchart). The quality assessment of these studies are displayed in Appendix A. The included systematic and narrative reviews are not discussed in the ‘results’ section since they did not include other relevant papers than the ones discussed below.

### 3.2. Description of Included Studies

The first RCT, by Denys et al. (2010) was a double-blind, sham–controlled, clinical trial of DBS of the nucleus accumbens (NA) that included 16 therapy resistant OCD patients [13]. In this study, refractoriness was defined as no or insufficient response to treatment with at least 2 different selective serotonin reuptake inhibitors (SSRIs) at maximum dosage for at least 12 weeks, treatment with clomipramine hydrochloride for at least 12 weeks with adequacy of treatment established by plasma levels, one augmentation trial with an atypical antipsychotic for 8 weeks in combination with an selective serotonin reuptake inhibitor, and at least 16 sessions of CBT [13].

Design: The study had 3 sequential treatment phases: an initial open phase, starting immediately after electrode implantation and lasting for 8 months, in which stimulation parameters were optimized and CBT was started. DBS was administered per protocol, with restricted stimulation settings at 90 µs, 130 Hz and a maximum stimulation intensity of 5.0 V. The effects of DBS were assessed with the Y-BOCS for obsessions and compulsions, with the Hamilton Anxiety Rating Scale (HAMA) for anxiety symptoms, and with the Hamilton Depression Scale (HAMD) for depression. After this open phase, a 1-month, double blind, sham-controlled phase started in which patients were randomly allocated to 2 periods of 2 weeks with the stimulators blindly turned ‘on’ (active stimulation) or ‘off’ (sham stimulation). CBT was continued throughout this phase. The double-blind phase was followed by a 12-month maintenance phase in which the stimulator was turned on for all patients and settings adjusted as required. Patients were allowed to use psychopharmacological medication during the trial.

Effectiveness: Stimulation in the initial open phase resulted in a mean decrease of 46% in Y-BOCS score from 33.7 (baseline) to 18.0 points; a mean decrease of 52% on the HAMA score from 20.9 (baseline) to 10.1 points, and a mean decrease of 46% in HAMD scores from 19.5 (baseline) to 10.5 points. Without stimulation, the improvement gained with the addition of CBT disappeared rapidly, suggesting that efficacy of CBT depends on stimulation. In this double-blind phase, the mean difference in Y-BOCS score between the active and sham condition after correction for period effects was 8.3 (*p* = 0.04). The mean difference in HAMA scores was 12.1 (*p* = 0.1) and the difference in HAMD scores was 11.3 (*p* = 0.01). It was reported that CBT was particularly effective in decreasing compulsions and avoidance behavior.

Mantione et al. (2014) performed a secondary analysis of this same RCT that was aimed at quantifying the added treatment effect of CBT after DBS, as well as to discuss the methodology of the CBT programme used (see above) [11]. The average decrease on the Y-BOCS after optimization of stimulation settings was 25%. With the addition of 24 weeks of CBT to ongoing DBS treatment, there was an additional 22% decrease of total Y-BOCS score (*p* = 0.021), without any additional effects on the HAMA or HAMD scores. The number of responders after CBT increased from 6 to 9 out of 16.

Timing: CBT was added when three conditions were fulfilled: an initial and substantial decrease (on average 6 points) in Y-BOCS score had to be obtained, there had to be no further decrease in Y-BOCS during three consecutive visits (which was usually after 8 weeks of stimulation), and it had to be observed that patients avoided resisting their compulsions or avoided anxiety-provoking exposure situations.

Procedural aspects: The CBT program consisted of 24 weekly individual face-to-face sessions of 60 min each. The protocolized treatment started with an extensive evaluation of the patient’s motivation. Once motivation was established, therapy started with ERP and gradually introduced more cognitive elements at later stages [14,21].

Tyaghi et al. (2019) performed a randomized, double-blind counterbalanced comparison of DBS of the anteromedian subthalamic nucleus (STN) and ventral capsule/ventral striatal (VC/VS) stimulation in 6 patients [12]. In this study, treatment resistance was defined as no sustained benefit from treatment with at least two SSRIs for a minimum of 12 weeks at optimal doses, augmentation of SSRI treatment with an antipsychotic or extension of the SSRI dose beyond recommended limits, and at least two trials of CBT with a minimum of 10 sessions, of which one as inpatient.

Design: the study consisted of two phases: an initial randomized phase of 12 weeks with stimulation of either the STN or the VC/VS, followed by an open phase in which both targets were stimulated. Stimulation was started after a mapping session at 60 µs and 130 Hz, without restrictions for stimulation intensity and also allowing stimulation of different contact points, both monopolar and bipolar. Next, there were two additional 12-week open phases: in the first one stimulation settings were optimized using data from previous phases. In the second phase, CBT was added to optimized DBS using the combined VC/VS and STN targets.

Effectiveness: Psychopharmacological treatment was allowed and kept constant during the trial. Overall, the score on the Y-BOCS reduced by 60%, from 36.2 at baseline to 14.3 when optimal stimulation settings were administered. Adding in-patient CBT resulted in an additional decline of the Y-BOCS score by 35% to 9.3 (*p* = 0.09; total decline from baseline 74%). Although obsessions and compulsions improved significantly from baseline to optimal stimulation, there was no statistically significant added improvement after CBT. Scores on the Montgomery–Asberg Depression Rating Scale (MADRS) declined from 28 at baseline to 13 during optimal stimulation settings and further reduced to 7 after CBT (information from the authors). The authors conclude that there is no further improvement in obsessions and compulsions due to CBT after optimal stimulation settings, and that this reflects a ‘floor effect’ of DBS on OCD. With ‘floor effect’ they intend to say that no further improvement of OCD symptoms would be possible after optimization of stimulation settings.

Timing: CBT was started standard after 24 weeks

Procedural aspects: CBT, including ERP, was applied in an inpatient unit while optimal stimulation settings were maintained.

Greenberg et al. (2006) report on the long-term (>3 years) follow-up of 10 therapy-resistant OCD patients being treated with VC/VS DBS [16].

Effectiveness: The average Y-BOCS score declined by 35% from 34.6 preoperatively to 22.3 after three years of DBS. All pharmacotherapy was allowed, but kept constant up to three months after the start of DBS treatment. The information provided in the paper does not allow to calculate the added effect of behaviour therapy to DBS on OCD symptoms. Clinically, the authors describe a ‘notably enhanced motivation to engage in goal directed activities’ during DBS, which also included enhanced motivation for CBT, which all patients had attempted unsuccessfully before the procedure. They consider that this increased motivation may have been a key factor in the patients’ clinical progress.

Timing: If patients had had behavior therapy immediately prior to DBS, this was allowed to continue after the start of DBS; if behavior therapy was newly indicated, it was allowed to start only six months after the start of DBS. No details on the number of patients that received behavioral therapy postoperatively is given, nor details about the type, frequency, duration and way of delivery of the behavioral therapy.

Procedural aspects: There is no information on procedural aspects.

Abelson et al. (2005) reported a case series of 4 therapy resistant OCD patients being treated with DBS of the anterior limb of the internal capsula, with the tip of the electrode adjacent to the nucleus accumbens [12]. After operation, a 12-week double-blind testing stage was followed by an open-ended, open stimulation phase, with efforts to optimize results by adjusting stimulation settings and by pharmacotherapy and CBT. No further details on the effectiveness, timing and procedural aspects of CBT is given.

The qualitative study by van Westen et al. (2019) reports on the results of interviews with 8 professionals involved in DBS treatment of OCD patients, as well as experiences from embedded patient observation of the author [17]. These professionals identified the process in which patients become increasingly engaged in their process of improvement as an important predictor of effect. As the patient changes, new possibilities emerge, one of which is renewed treatment with CBT, to reduce remaining symptoms and expand healthy behavioral repertoires [17].

## 4. Discussion

Whereas it is a common practice to offer patients with therapy-resistant OCD treated with DBS a course of CBT after their operation, its effectiveness, timing and procedural aspects, such as the preferred way of delivery of such therapy, has hardly been studied. In spite of the fact that the importance of post-operative CBT is stressed by various authors [18,19,20], only two trials have specifically focussed on CBT added to DBS [13,14,15].

### 4.1. Effectiveness

The two studies that assessed the effects of CBT added on to DBS both support its effectiveness. In the study by Denys et al., CBT was responsible for significant additional reduction of 22% on the Y-BOCS after optimal stimulation settings were achieved [14]. In the study by Tyagi et al., there was a trend for an additional improvement of 35% on the Y-BOCS (*p* = 0.09). Whereas this was not statistically significant, there is a clear trend towards significance for this finding and it constitutes a clinically relevant improvement. In our opinion the lack of a statistical significance may well be due to a power problem because of the very low number of included patients (*n* = 6). So contrary to the authors, who present this as a negative outcome, we consider this study in support of postoperative CBT.

In theory, the effectiveness of CBT may also depend on the preoperative cognitive state of the patient, as well as on the potential cognitive side effects of DBS. Whereas in patients with Parkinson’s disease, cognitive side effects of DBS—especially of the subthalamic nucleus—have been associated with reduced processing speed and working memory [22], there is little evidence of any detrimental effect of DBS—of any target—on the cognitive performance of OCD patients [23]. Studies that do report on neuropsychological measures report no relevant change in cognitive performance after DBS [24], and in one case even an improvement in cognitive flexibility for STN DBS but not for VC/VS DBS [15]. None of the included papers report on problems administering CBT due to cognitive side effects.

There has been some discussion on whether the effects of CBT may depend on the DBS target. Mantione et al. suggest that the effect of CBT in their study may be specific to stimulation of the NA, since NA DBS has a profound effect on anxiety and depression, as opposed to, e.g., DBS of the STN, which reduces compulsions without significant effects on mood and anxiety [11]. However, in the study by Tyagi et al., the additional improvement in patients with STN and VC/VS DBS is in the same range, if not larger than in the study by Mantione et al. [15]. Based on these scarce data, we expect CBT to be effective as add-on treatment to DBS in therapy-resistant OCD patients, irrespective of the stimulation target. However, only further studies comparing the effectiveness of CBT in OCD patients with different DBS stimulation targets can reveal a potential target related effect of CBT.

### 4.2. Timing

The same two studies used different criteria for starting CBT. Denys et al. started CBT after partial response, defined as a substantial reduction of on average 6 points on the Y-BOCS, without further decrease during three consecutive visits [13,14]. In the study by Tyagi, CBT was started after 36 weeks in every patient, irrespective of the amount of improvement achieved by DBS [15]. In clinical practice the question of when to best start CBT is an important one. It does not seem sensible to start CBT immediately postoperatively after electrode implantation and activating the DBS system. The mental state of the patient has not changed yet, and because of that there is no reason to expect that treatments that were ineffective before DBS would now be effective. It also makes no sense to start CBT if the response to DBS is very large, since there may not be any relevant treatment goals left to work towards. The best time to start CBT is probably when there is a partial response to DBS. The altered mental state of the patient, with not only some reduction in obsessive-compulsive behavior, but usually also reduced anxiety and improved mood, provides a different starting position for CBT, and the patient may be more able and motivated to comply with therapy, as was also described by Greenberg et al., and by Van Westen et al. in their qualitative study [16,17]. The question then is when to start CBT in case of partial response? Some clinicians routinely start after a certain period (e.g., after 8 or 12 weeks). Other clinicians start CBT when it is assumed that optimal stimulation settings have been achieved. Whereas this may be preferable in a research context, in order to separate the differential contribution of DBS and CBT to the response, clinically this is debatable. On one hand, reaching optimal stimulation settings may take a long time in many patients, which would lead to an unacceptable delay for therapy and loss of momentum; on the other hand, these patients have already experienced non-effective CBT and it is important to spare them another failure because of starting CBT to soon, as this would decrease their motivation for another attempt when stimulation parameters are optimal. One option is to assess the “readiness” for CBT, as mentioned above. Another option may be to look for improvement of cognitive measures that may increase the likelihood of successful CBT. A recent intervention study showed that the effect of VC/VS DBS is explained in part by enhancement of cognitive control by the prefrontal cortex. In this study, DBS improved the patients’ performance on a cognitive control task and increases theta (5–8 Hz) oscillations in both medial and lateral PFC, which predicts the clinical outcome [25]. Perhaps such indicators could be made to clinical use and help to indicate the best time to start CBT after DBS.

### 4.3. Procedural Aspects

In the study by Denys et al., CBT/ERP consisted of 24 weekly individual face-to-face sessions of 60 min each, administered on an outpatient basis. Tyagi et al. provided CBT/ERP for 12 weeks on an in-patient basis in a neuropsychiatry unit. In both studies, the therapy was provided by the DBS clinic. This may be feasible in a research setting, but in routine clinical practice this will be more difficult to ask from patients once the DBS settings are optimized, given the distance that many of them will have to travel to the DBS clinic. Because of this, therapy is often organized in the region where the patient lives. However, whereas many behavioral therapists from local or regional psychiatric services may have experience in treating OCD patients, few will have experience treating OCD patients with DBS. In-patient treatment is one option to let patients benefit from the expertise of therapists of the DBS clinic, but this will be costly and may not be more effective than out-patient treatment. Another way of letting patients benefit from therapists with DBS experience is to explore novel ways of administering therapy, such as by telephone, videoconferencing or online.

In addition, other indications for CBT in the peri-operative period should also be considered. CBT could be administered with different objectives and if necessary, a different procedural approach. It could for instance already be started pre-operatively with the intent to enhance motivation for change post-operatively. Such pre-operative intervention has not been studied yet. Additionally, the content of the cognitive aspects of therapy could be adapted to address some issues specific to DBS, such as specific psychoeducational purposes related to DBS and preoccupation with stimulation settings. Moreover, after substantial improvement, low frequency long term continuation therapy may be helpful in preventing relapse.

### 4.4. Synthesis and Recommendations

Only two studies specifically address postoperative CBT. These used different stimulation targets and stimulation protocols, as well as different approaches to administering the therapy. Both studies suffer from a number of limitations, most importantly a small sample size, and the lack of a control condition for the CBT. In addition, the focus is strongly on obsessive and compulsive symptoms, whereas a focus on quality of life and general (social) functioning may be more important to the patient [26]. The other included studies mention postoperative CBT, but do not provide any details on effectiveness, timing and procedure.

DBS is not a stand-alone treatment for therapy-resistant OCD. After their operation, many patients continue to take medication for OCD, and/or receive some form of psychotherapy to deal with remaining symptoms or problems adjusting to the new situation. The overall treatment effect is the resultant of the DBS plus adjunctive therapies, and studies into the effectiveness of DBS should also take these concurrent treatments into account.

From a clinical point of view, there is a need for an evidence based algorithm for applying concomittant therapies, both psychotherapy as well as pharmacotherapy. As far as psychotherapy is concerned, there should be clear criteria as to when to start psychotherapy and the module should be adjusted to patients being treated with DBS. In our opinion, CBT should be started after a predefined level of clinical response to DBS, which is open for dicussion. The CBT module should address issues specific to DBS patients such as a changed personal identity due to being dependent on a device for symptom control and well-being, preoccupation with stimulation settings, and having to adjust to ‘real life’ after a long time of therapy-resistance and severe obsessive-compulsive behaviors that rendered typical family life, social contacts or employments unfeasible [27]. In order to let patients benefit from the experience of CBT therapists working in DBS clinics, other ways of administering CBT such as by telephone, videoconferencing or online, should also be developed and evaluated.

From a research point of view, future studies into the efficacy of DBS for OCD should follow a design that also allows the evaluation of the added effect of these concurrent treatments, and helps determining the place of these concurrent treatments in a treatment algorithm of OCD patients after DBS. This implies that there should be a control condition for CBT in order to assess the placebo response of CBT treatment. It also implies that sample size should be large enough to allow evaluation of the added treatment effects of CBT. Since it is unlikely that the required sample sizes will be achieved within a reasonable amount of time in a single DBS center, multicenter studies should be initiated. It is essential that collaborating centers not only protocolize their CBT treatment, but also that they align their clinical practice with respect to DBS with respect to stimulation target, strategies to optimalize stimulation parameters and follow-up assessments. This would require a closer collaboration between DBS clinics on both a national and international level.

## 5. Conclusions

Preliminary findings show that postoperative CBT is effective as an add-on treatment to DBS in patients with therapy-resistent OCD. Further studies are necessary to establish the place of CBT after DBS. These studies should have larger sample sizes and designs that are adequate to quantify the added effects of both CBT as well as pharmacotherapy. In order to let patients benefit optimally from the experience and expertise of behavioral therapists working in DBS clinics, novel ways of administering CBT, such as administered by telephone, videoconferencing or online, should also be studied.

## Figures and Tables

**Figure 1 jcm-09-02383-f001:**
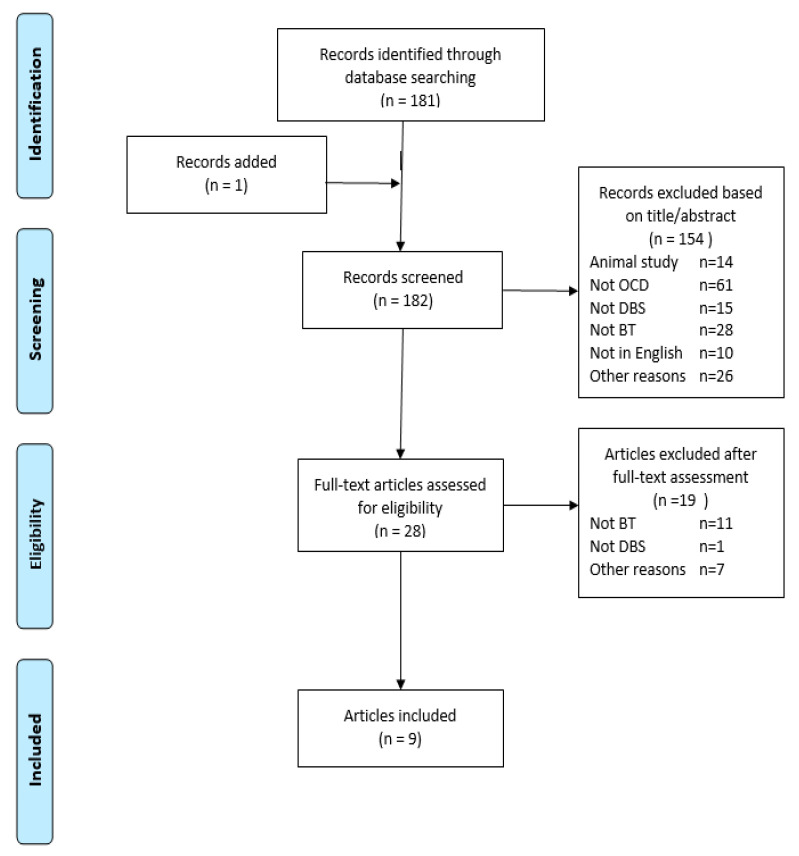
PRISMA flow diagramme. Abbreviations: OCD = obsessive compulsive disorder; DBS = deep brain stimulation; BT = behavioral therapy.

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
