# Peer review of "Effectiveness, Timing and Procedural Aspects of Cognitive Behavioral Therapy after Deep Brain Stimulation for Therapy-Resistant Obsessive Compulsive Disorder: A Systematic Review"

_jcm, 2020, doi:10.3390/jcm9082383_

Round 1

Reviewer 1 Report

This is clearly a little studied area but one that is very significant for those who suffer from severe OCD. The literature section need be improved.

I think the authors could do 2 things to make this manuscript more useful.

  1. make some comments about DBS location - there was an assumption that DBS works yet the target locations were different across studies.  has there been enough literature to comment on this as this is the basis for then looking at the effects of CBT + DBS.
  2. Given the literature needs to be improved, I would like to see the authors make some statements in the discussion about some key things that need to be considered in future studies such as 1) what outcomes should be examined uniformly across studies, 2) something about elibility criteria, 3) discussion about basic study design principals to incorporate - some type of comparison group (maybe not randomized but there are other methods of developing a comparison), timing of measurement, reduction of confounds, etc.  One goal of this paper could be to point readers to do studies that will help us to answer the question about the effects of DBS + CBT in OCD.

Author Response

This is clearly a little studied area but one that is very significant for those who suffer from severe OCD. The literature section need be improved.

We thank thereviewer for acknowledging the significance of the topic.

I think the authors could do 2 things to make this manuscript more useful.

1. make some comments about DBS location - there was an assumption that DBS works yet the target locations were different across studies.  has there been enough literature to comment on this as this is the basis for then looking at the effects of CBT + DBS.

Answer: the effect of stimulation target was already briefly discussed on page 6. The text was slightly adjusted in order to make this point more explicit (page 6)

2. Given the literature needs to be improved, I would like to see the authors make some statements in the discussion about some key things that need to be considered in future studies such as 1) what outcomes should be examined uniformly across studies, 2) something about elibility criteria, 3) discussion about basic study design principals to incorporate - some type of comparison group (maybe not randomized but there are other methods of developing a comparison), timing of measurement, reduction of confounds, etc.  One goal of this paper could be to point readers to do studies that will help us to answer the question about the effects of DBS + CBT in OCD.

Answer: in the discussion section the subsection 'synthesis' was altered into 'synthesis and recommendations', and two paragraphs were added, one with recommendations for clinical practice and one with recommendations for research (pages 7 and 8)

Reviewer 2 Report

The Topic is very interesting and attractive because we do not have many reviews in the area of " OCD and DBS". However, the big concern is the total number of RCTs is very limited ( only 2-3 RCT) and based on the limited number of RCT studies, this review might not fully reflect the true picture of "OCD and DBS". It is better to wait until we have sufficient RCTs to write such a systematic review.

Author Response

The Topic is very interesting and attractive because we do not have many reviews in the area of " OCD and DBS".

Answer: we thank the reviewer for acknowledging the interest of this topic. The authors want to stress that there is not a single review on the added effect of CBT to DBS for OCD.

However, the big concern is the total number of RCTs is very limited ( only 2-3 RCT) and based on the limited number of RCT studies, this review might not fully reflect the true picture of "OCD and DBS". It is better to wait until we have sufficient RCTs to write such a systematic review.

Answer: We set out on performing a systematic review with the aim to use this information for reviewing and organizing our own policy regarding postoperative CBT. I agree the yield of the literature search was disappointingly limited. We did however follow the PRISMA guidance for systematic reviews and prefer to report it like that. Reporting the findings as a systematic review (which in fact it is) will draw more explicit attention to this gap in our knowledge.

Reviewer 3 Report

The authors conducted a systematic review of studies cataloged in PubMed following the Preferred Reporting Items for Systematic Reviews and Meta-Analyses (PRISMA) guidelines. Eventually, 9 papers were included in the review: three randomized controlled trials (RCTs), one cohort study, one case series, one qualitative study, one systematic review and 2 narrative reviews. Two of the included papers were based on the same RCT. Although the initial intention was to perform a meta-analysis, this was not possible because of the limited number of included studies that, in addition, used different indication criteria and different forms of CBT. Timing and procedural aspects of CBT in the studies are reported in a descriptive way. I have some major comments for this review.

1. The authors should clarify what are the new and adding findings compared to several preceding reviews.

2. Please add the definition of treatment resistant OCD for each RCT. If participants failed to apply CBT before introducing DBS, applications of CBT after DBS may be the sign of efficacy of DBS.

3. Because cognitive impairments by DBS in Parkinson’s disease were well known and confirmed by meta-analyses, the possible influences of cognitive impairments on efficacy of CBT should be discussed.

4. Please add brief information on the guidelines of each country for DBS for refractory OCD. In my knowledge, some contries approve specific DBS for OCD but other contries not. What types of DBS are the recommended for refractory OCD?

Author Response

The authors conducted a systematic review of studies cataloged in PubMed following the Preferred Reporting Items for Systematic Reviews and Meta-Analyses (PRISMA) guidelines. Eventually, 9 papers were included in the review: three randomized controlled trials (RCTs), one cohort study, one case series, one qualitative study, one systematic review and 2 narrative reviews. Two of the included papers were based on the same RCT. Although the initial intention was to perform a meta-analysis, this was not possible because of the limited number of included studies that, in addition, used different indication criteria and different forms of CBT. Timing and procedural aspects of CBT in the studies are reported in a descriptive way.

Thank you.

I have some major comments for this review.

1. The authors should clarify what are the new and adding findings compared to several preceding reviews.

Answer: there are no previous reviews on the added effect of CBT to DBS in OCD patients. The included reviews were reviews concerning the effect of DBS in OCD, and did not report on quantitative added effects of CBT. These reviews were still included if they provided any comments or opinion on postoperative CBT. I our revision this is now more explicitly stated on pages 2 and 3.

2. Please add the definition of treatment resistant OCD for each RCT. If participants failed to apply CBT before introducing DBS, applications of CBT after DBS may be the sign of efficacy of DBS.

Answer: the criteria for refractoriness are now mentioned for both included studies. Both studies required a failed course of CBT prior to surgery (page 4)

3. Because cognitive impairments by DBS in Parkinson’s disease were well known and confirmed by meta-analyses, the possible influences of cognitive impairments on efficacy of CBT should be discussed.

Answer: there are no clear indications of altered cognition in OCD patients treated with DBS. The issue is now explicitly addressed in a new paragraph in the discussion section (page 6)

4. Please add brief information on the guidelines of each country for DBS for refractory OCD. In my knowledge, some countries approve specific DBS for OCD but other countries not. What types of DBS are the recommended for refractory OCD?

Anwer: there are no specific targets specified in the EC or FDA approval of DBS for OCD. One line was inserted in the introduction referring to EC and FDA approval. We considered going into specific requirements per country to be beyond the scope of this paper.

Round 2

Reviewer 1 Report

Authors were responsive to all reviewer comments.  no additional suggestions.

Author Response

Thank you. We made some minor textual adaptations.

Reviewer 2 Report

The topic is very interesting and meaningful even the RCT number of current studies is limited. The authors have tried very hard to answer all questions from the reviewers and did make the  improvement and make the manuscript more publishable. 

Actually, the author did their best to answer all questions, please double check the language and any typos. No more suggestion due to the limited published current data.

Author Response

(The authors gave the same response as above.)

Reviewer 3 Report

The authors successfully incorprated the reviewr's comments into the revised manuscript.

Author Response

Thank you